# The Effect of Subjective Exercise Experience on Exercise Behavior and Amount of Exercise in Children and Adolescents: The Mediating Effect of Exercise Commitment

**DOI:** 10.3390/ijerph191710829

**Published:** 2022-08-30

**Authors:** Linghui He, Yan Li, Zhenhuai Chen

**Affiliations:** 1Jinshan Elementary School, Liangjiang New District, Chongqing 401122, China; 2Humanities and Sports Research Center, Southwest University, Beibei District, Chongqing 400715, China; 3Sports Work Department, College of Liberal Studies, Chongqing Industry Polytechnic College, Yubei District, Chongqing 401120, China; 4Faculty of Physical Education, China West Normal University, Nanchong 637009, China

**Keywords:** children and adolescents, subjective exercise experience, exercise commitment, exercise behavior, mediating effect

## Abstract

Purpose: To explore the influencing factors that restrict the exercise behavior of children and adolescents, investigate the effect of subjective exercise experience on exercise behavior, and reveal the mediating effect of exercise commitment between subjective exercise experience and exercise behavior so as to promote children and adolescents to maintain good health exercise habits and improve their physical and mental health. Methods: The Subjective Exercise Experience Scale (SEES), Exercise Commitment Scale (ECC), and Physical Exercise Rating Scale (PARS-3) were used to conduct a questionnaire survey on 600 children and adolescents in Chongqing, China, and SPSS21.0 and AMOS21.0 statistical analysis software was used to carry out statistics and analyses on the questionnaires. Results: (1) Among children and adolescents, boys’ exercise commitment and exercise behavior were significantly higher than girls’, and there was no significant gender difference in subjective exercise experience. The exercise behavior of children and adolescents aged 9–12 was significantly higher than that of children and adolescents aged 13–15, and there was no significant age difference in subjective exercise experience and exercise commitment. (2) There was a significant correlation between the subjective exercise experience, exercise commitment, and exercise behavior of children and adolescents, and subjective exercise experience could directly and positively predict exercise commitment (*β* = 0.63) and exercise behavior (*β* = 0.57)—exercise commitment could also directly and positively predict exercise behavior (*β* = 0.52). (3) The exercise commitment of children and adolescents has a partial mediating effect between subjective exercise experience and exercise behavior (accounting for 37.50% of the total effect), and has a mediating effect between different exercise amounts, with the strongest mediating effect being on high exercise amount (32.10% of the total effect). Conclusions: The exercise behavior of children and adolescents was not only directly affected by subjective exercise experience, but also affected by the mediating effect of exercise commitment, and maintaining a good exercise experience and commitment was an effective way to effectively improve exercise behavior and amount of exercise in children and adolescents.

## 1. Introduction

According to the results of the 2018 Student Physical Health Test, although the physical health status of Chinese children and adolescents has improved, the obesity rate and myopia rate were still high, and children and adolescents have not formed good sports awareness and exercise behavior [1]. Studies have shown that when children and adolescents lack sufficient physical awareness and exercise behavior, it was easy for them to have less independent physical exercise in addition to the physical activities prescribed by the school, which in turn leads to a decline in their physical fitness and physical health [2,3]. Conversely, regular exercise behavior could have many powerful effects on the physical and mental health of children and adolescents [4,5]. For example, active physical activity behaviors have been shown to improve children and adolescents’ mental health [6,7], improve academic performance [8], improve cognitive function [9], and reduce the risk of obesity and cardiovascular disease in children and adolescents [10]. Of course, ideal exercise habits should be reflected in regular exercise frequency and exercise duration, being maintained over a relatively long period [11,12,13]. It can be seen that developing good exercise behaviors and maintaining sufficient exercise were crucial to promoting the physical and mental health of children and adolescents. Then, finding out the factors that restrict or develop exercise behaviors may be an important way to promote children and adolescents to develop good exercise behaviors.

An individual’s exercise behavior was affected by both internal factors (such as psychology) and external factors (such as the environment) [14,15], and internal factors were important internal driving forces for maintaining, changing, and developing exercise behavior. The social learning theory holds that experiences in specific situations could promote adaptive changes in individual behavioral perception, decision-making, and expression that determine behaviors [16,17]; among them, as an important antecedent variable, the subjective exercise experience was considered to be an irrational influencing factor on individual exercise behavior [18]. This refers to the individual’s feelings and impressions of past exercise emotions, and it is the self-assessment of the exercisers’ experience of a positive/negative emotional state and physiological consumption after exercise [19], wherein a positive exercise experience could be internalized into internal motivation, forming exercise intention for the purpose of pursuing fun—accompanied by the desire and determination to practice exercise repeatedly [20]; however, those who lack a positive experience or had a negative exercise experience and consistent negative cognitive responses to exercise activities, often show rejection and resistance tendencies, which severely restrict exercise behavior [21,22]. Research shows that subjective exercise experience was not only a cognitive operation experience [19], but also a behavioral fluency state [23], which may play an important predictive role in individual exercise behavior. For children and adolescents, they were in a critical period of cognitive and behavioral formation and rapid development. Therefore, when they have positive cognition and a positive experience of their own exercise experience or behavior in the past, they could enhance their exercise intention and interest, and may be more likely to produce positive and lasting exercise behavior. On the contrary, it was easy to have corresponding resistance to exercise behavior, thereby restricting the occurrence of exercise behavior. Meanwhile, the rational psychology of individuals will determine the consolidation and stability of behavioral habits [24], and the maintenance and development of exercise behaviors were closely related to rational psychological factors. Among them, commitment was a psychological contract for individuals to practice exercise based on cognition, and it was shown that people who can put their emotions into physical exercise were more likely to show a firm determination to exercise—and then they will form regular exercise behaviors [25,26]. Studies have pointed out that exercise commitment, as a rational psychological state, was a psychological motivation for the desire and determination to continue to participate in exercise [27]. Exercise commitment was a powerful factor for maintaining exercise adherence and avoiding the interruption of exercise behavior, which could effectively reflect individual exercise intention and motivational intensity [28], i.e., the higher the exercise commitment level of exercisers, the better their exercise behavior adherence will be—and the less likely they will be to withdraw from exercise [29]. Good exercise commitment may be another important predictor of exercise behavior in children and adolescents, but there is currently a lack of direct evidence on the path effect of the relationship between the two variables.

Exercise psychology believes that people tend to formulate behavioral plans, evaluate future events, and decide whether to perform behaviors based on existing behavioral experiences [30]. Therefore, exercise commitment was considered to be an after-effect of subjective exercise experience [31], and the subjective exercise experience could effectively predict individual exercise persistence, and exercise commitment plays a significant mediating role between the two [32]. That is to say, a positive exercise experience will help to enhance an individual’s exercise intention and strengthen their determination to persist in exercising, while a negative exercise experience will inhibit exercise intention, weaken motivation, and restrict plan implementation, resulting in a tendency to withdraw [33,34], which in turn may limit the occurrence of exercise behaviors. Therefore, in children and adolescents, subjective exercise experience and exercise commitment may both be important predictors of developing or restricting their exercise behavior, but the specific path relationship among the three variables needs to be further explored.

To sum up, the exercise behavior of children and adolescents is of great significance to promote their physical and mental health. However, individual exercise behavior was easily regulated by rational and irrational psychological factors, among which there was a relatively close path relationship between subjective exercise experience, exercise commitment, and exercise behavior. However, in children and adolescents, key issues, such as the effect of subjective exercise experience on improving exercise behavior and exercise amount and the role of exercise commitment in this, remain unclear. In view of this, this study proposes the following hypothesis: (H1) subjective exercise experience has a direct predictive effect on children and adolescents’ exercise behavior; (H2) subjective exercise experience has a direct predictive effect on children’s and adolescents’ exercise commitment; (H3) exercise commitment has a mediating effect between children and adolescents’ subjective exercise experience and exercise behavior; and (H4) there were differences in the mediating effect of exercise commitment between subjective exercise experience and exercise behavior with different exercise amounts.

## 2. Participants and Methods

### 2.1. Participants

Using a cross-sectional survey research design, this study used a cluster random sampling method to conduct a sample survey among 7 primary and secondary schools in Chongqing, China, with a sampling ratio of approximately 1:40 in each school. The selected students were informed about participating in this research and were assigned to a designated classroom to complete the questionnaire. The inclusion criteria were: (1) Aged 9–15 years, (2) full-time students, and (3) voluntary participation in this study. The exclusion criteria were: (1) People with intellectual disabilities and (2) people with mental disabilities. To ensure the authenticity of the questionnaires, we fully explained the contents and precautions of the questionnaires to the participants before issuing the questionnaires and adopted the method of distributing and returning the questionnaires on site. The participants needed to spend 18 min in class to complete the questionnaires. In this study, 80 to 100 students were randomly selected as subjects in each school sampled, and a total of about 600 questionnaires were distributed, with a total of 568 being returned. Through screening (exclusion principle: missing key content information and regular answering, etc.), 540 valid questionnaires were determined (95.07% effective rate). Among them, there were 232 boys and 308 girls, the average age was (13.17 ± 3.54) years, and the average BMI was (20.52 ± 4.88). This study was approved by the Ethics Review Committee of the School of Physical Education, Southwest University (SWU-TY202105), and followed the Declaration of Helsinki. Furthermore, written informed consent was obtained from all participants. In addition, since the survey object is a group of minors, we also obtained the approval of the parents of the participants, who signed the parental informed consent.

### 2.2. Measuring Tools

#### 2.2.1. Subjective Exercise Experience Scale (SEES)

The two subscales of “positive well-being” and “psychological distress” in the “Subjective Exercise Experience Scale” compiled and revised by Mcauley [35] were selected (each containing 4 items, with a total of 8 items), and the Likert 5-point scoring scale was adopted. The options “completely incompatible–completely in line” were counted as 1–5 points, respectively. Considering that the “psychological distress” scale was an assessment indicator for expressing negative experiences, the sum of the scores of each item of “positive well-being” was used to evaluate the workout experience level of the participants, after the reverse processing of each item of “psychological distress”, where a higher score indicated a better subjective exercise experience. After the test-retest, it was found that the item loads of the “positive well-being” and “psychological distress” scales were all between 0.50 and 0.95, the combined reliability (CR) was greater than 0.60, and the average variance extraction value (AVE) was also greater than 0.50. It shows that the convergent validity of the two scales was at a good level, and the AVE values of each scale were all greater than the square value of the correlation coefficient, indicating that they have good discriminant validity. After the internal consistency test, the Cronbach α coefficients of “positive well-being” and “psychological distress” were 0.85 and 0.82, respectively, and the overall Cronbach α coefficient of the total scale was 0.84. The confirmatory factor analysis was performed on the scale, and the results of the measurement model validation were: x^2^/df = 2.05, TLI = 0.97, CFI = 0.98, AGFI = 0.95, IFI = 0.93, RMSEA = 0.04. This shows that the scale has good construct validity and reliability.

#### 2.2.2. Exercise Commitment Checklist (ECC)

The Exercise Commitment Scale, which was revised by Chen et al. [36], was adopted. It includes 5 dimensions of exercise commitment, exercise fun, personal engagement, social restraint, and participation opportunity (3 items for each dimension, 15 items in total). Using Likert 5-point scoring, according to the options “strongly disagree”–“strongly agree” were counted as 1–5 points, respectively, and the sum of the five dimensions was the exercise commitment level of the subjects in the physical exercise situation; furthermore, the higher the total score, the higher the exercise commitment level. After the test-retest, it was found that the load of each item in the 5 dimensions was between 0.50 and 0.95, the combined reliability (CR) was greater than 0.60, and the average variance extraction value (AVE) was also greater than 0.50. This shows that the convergent validity of the five dimensions was good, and the AVE value of each dimension was all greater than the square value of the correlation coefficient, indicating that they all have good discriminant validity. After the internal consistency test, the Cronbach α coefficients of exercise commitment, exercise fun, personal investment, social constraints, and participation opportunities were 0.86, 0.78, 0.80, 0.83, and 0.80, respectively, and the overall Cronbach α coefficient of the total scale was 0.82. The confirmatory factor analysis was performed on the scale, and the results of the measurement model validation were: x^2^/df = 1.88, TLI = 0.99, CFI = 0.95, AGFI = 0.96, IFI = 0.94, RMSEA = 0.03. This shows that the scale has good construct validity and reliability.

#### 2.2.3. Physical Activity Rating Scale (PARS-3)

The Physical Activity Rating Scale of Liang [37] was adopted and revised to evaluate three aspects, including exercise intensity, exercise frequency, and exercise time. Using the Likert 5-point scoring, the exercise intensity and frequency were scored from 1 to 5 on a scale of 1 to 5, and the exercise time was scored from 0 to 4 on a scale of 1 to 5; the formula, “exercise intensity × exercise time × exercise frequency”, was used to quantify the total score of the subjects’ exercise behavior (the lowest score was 0 points and the highest score was 100 points), and a higher score means a greater amount of exercise. At the same time, the evaluation criteria for exercise amount were: low exercise amount ≤19 points, moderate exercise amount 20–42 points, and high exercise amount ≥43 points. The test-retest reliability of the scale was high, and the correlation coefficient, r = 0.82.

The factor extraction and reliability analysis of the three measurement scales are shown in Table 1.

### 2.3. Data Analysis

This study used SPSS21.0 software (IBM, Armonk, NY, USA) to process and analyze the data, and used a factor analysis, test-retest method, and internal consistency test to examine the reliability and validity of the scale. The descriptive statistics and an independent sample *t*-test were used to investigate the status quo of children and adolescents’ subjective exercise experience, exercise commitment, and exercise behavior. The Pearson correlation analysis, regression analysis, bootstrap analysis, and the use of AMOS21.0 software (IBM, Armonk, NY, USA) to establish a structural equation model were used to investigate the relationship between the variables and the mediating effect of exercise behavior. The significance level of all indicators was set at α = 0.05.

## 3. Results

### 3.1. Common Method Bias Test

Since the questionnaire survey method was used in this study, all the questionnaire items were filled out by the subjects themselves, so there may be a common method bias in the measurement. Therefore, in order to minimize the influence of the common method bias on the results, this study adopted procedure control methods, such as anonymous questionnaire measurement and standardized test administration, to control it accordingly. Before the test, the instructions were explained, the subjects were guaranteed that their participation was voluntary, and the data were collected by the method of group testing and were collected on the spot after completing the test. After the data collection was completed, the Harman one-factor test was used to investigate the problem of common method bias [38], which was an unrotated factor analysis that puts the measurement items of all variables together, builds a one-factor model, and then compares the fit indices of the one-factor models. The results showed that there were 8 factors with eigenvalues greater than 1, and the variance explained by the first factor was 26.32%, which was less than the critical standard of 40%, indicating that the common method bias does not cause serious problems in this study.

### 3.2. Demographic Differences in Primary Variables

The independent sample *t*-test found that in terms of gender, boys’ exercise commitment (*t* = 3.55, *p* < 0.05) and exercise behavior (*t* = 3.37, *p* < 0.05) were significantly higher than girls’, but there were no significant gender differences in subjective exercise experience (*t* = 1.52, *p* > 0.05). In terms of age, the exercise behavior of children and adolescents aged 9–12 was significantly higher than that of children and adolescents aged 13–15 (*t* = 3.21, *p* < 0.05), however, there was no significant age difference in subjective exercise experience (*t* = 0.87, *p* > 0.05) and exercise commitment (*t* = 2.45, *p* > 0.05). (see Table 2).

### 3.3. Correlations among Children and Adolescents’ Subjective Exercise Experience, Exercise Commitment, and Exercise Behavior

The Pearson correlation analysis found that the subjective exercise experience of children and adolescents was significantly positively correlated with exercise commitment (*r* = 0.63, *p* < 0.001) and was significantly positively correlated with exercise behavior (*r* = 0.57, *p* < 0.001), and exercise commitment was significantly positively correlated with exercise behavior (*r* = 0.52, *p* < 0.001). The correlations among the main variables reached a significant level, which provided a good basis for the subsequent test of the mediation effect (see Table 3).

### 3.4. The Effect and Path Relationship of Subjective Exercise Experience on Exercise Behavior

#### 3.4.1. The Direct Effect Analysis

The linear regression analysis was used to examine the direct relationship between the variables (Table 4). First, after controlling for demographic variables, such as gender and age, this study used subjective exercise experience as an independent variable and exercise commitment and exercise behavior as dependent variables, respectively. The results showed that the subjective exercise experience could positively predict the exercise commitment of children and adolescents (*β* = 0.63, *p* < 0.001), which could explain 39.69% of the variance; furthermore, the subjective exercise experience also positively predicted exercise behavior (*β* = 0.57, *p* < 0.001), explaining 32.49% of the variance. Secondly, using exercise commitment as an independent variable and exercise behavior as a dependent variable, the exercise commitment could positively predict individual exercise behavior (*β* = 0.52, *p* < 0.001), which could explain 27.04% of the variance.

#### 3.4.2. Analysis of Mediation Effect

This study draws on the mediation effect test method of Baron and Kenny [39], which mainly includes three steps: First, the independent variable has an influence on the dependent variable, and the regression coefficient was significant; second, the independent variable has an influence on the intermediary variable, and the regression coefficient was significant; and third, the common influence of the independent variable and the intermediary variable on the dependent variable reaches a significant level, and the influence of the intermediary variable on the dependent variable must reach a significant level at this time. If the influence of the independent variable on the dependent variable becomes insignificant, then the mediator variable plays a complete mediating role. Furthermore, if the influence of the independent variable on the dependent variable was reduced but still significant, the mediator variable plays a partial mediating role.

The AMOS software was used to establish a structural equation model to investigate the mediating effect of exercise commitment between children and adolescents’ subjective exercise experience and exercise behavior. The fitting indicators of the model were: x^2^/df = 1.96, RMSEA = 0.02, GFI = 0.96, TLI = 0.99, NFI = 0.94, IFI = 0.91, AGFI = 0.95, indicating that the model has a good degree of fit and is suitable for the mediation effect test. The model results (Figure 1) showed that the path coefficient of subjective exercise experience on exercise behavior was significant (*β1* = 0.57, *SE* = 0.02, *p* < 0.001). After adding exercise commitment as a mediating variable, the path coefficient of subjective exercise experience to exercise commitment was significant (*β* = 0.63, *SE* = 0.04, *p* < 0.001), and the path coefficient of exercise commitment to exercise behavior was significant (*β* = 0.47, *SE* = 0.03), *p* < 0.001). However, the path coefficient of subjective exercise experience on exercise behavior decreased, but still reached a significant level (*β2* = 0.50, *SE* = 0.03, *p* < 0.001), indicating that exercise commitment played a partial mediating role between subjective exercise experience and exercise behavior (the effect decomposition of each path is shown in Table 5). Therefore, the hypotheses H1, H2, and H3 of this study were all confirmed.

The results showed that children and adolescents’ exercise behavior was not only directly affected by subjective exercise experience, but also affected by the mediating effect of exercise commitment, while exercise behavior was a comprehensive variable, and there was often a “dose effect” of exercise amount. Given this, does exercise commitment partially mediate between subjective exercise experience and different exercise amount? Fang et al. [40] found that the bias-corrected percentile Bootstrap method (Bias-corrected Bootstrap Method) was more effective than the traditional Sobel test when doing the significance test of the mediation effect. In view of this, in order to further examine the mediating effect of children and adolescents’ exercise commitment between subjective exercise experience and different amounts of physical exercise, this study adopted Model 4 of the SPSS macro compiled by Hayes [41], which estimates the 95% confidence interval of the mediation effect by taking 5000 samples and conducts the mediation effect test of different physical exercise amounts, respectively. If the 95% confidence interval of the mediation effect does not include 0, it means that the mediation effect was significant; otherwise, it means that the mediation effect was not significant. The mediation effect test in this study was conducted under the control of statistical variables, such as gender and age.

The regression analysis results showed (Table 6, Table 7 and Table 8): (1) The subjective exercise experience could significantly and positively predict exercise commitment (*β* = 0.39, *p* < 0.01), and exercise commitment could significantly and positively predict low exercise amount (*β* = 0.22, *p* < 0.05); when both predict low exercise amount at the same time, the former could significantly and positively predict low exercise amount (*β* = 33, *p* < 0.01). (2) The subjective exercise experience could significantly and positively predict exercise commitment (*β* = 0.49, *p* < 0.001), and exercise commitment could significantly and positively predict moderate exercise amount (*β* = 0.41, *p* < 0.01); when both predict moderate exercise amount at the same time, the former could significantly and positively predict moderate exercise amount (*β* = 34, *p* < 0.01). (3) The subjective exercise experience could significantly and positively predict exercise commitment (*β* = 0.61, *p* < 0.001), and exercise commitment could significantly and positively predict high exercise amount (*β* = 0.55, *p* < 0.001); when both predict high exercise amount at the same time, the former could significantly and positively predict high exercise amount (*β* = 43, *p* < 0.001).

Figure 2, Figure 3 and Figure 4 showed the path model and coefficient values of children and adolescents’ exercise commitment between subjective exercise experience and different physical exercise amounts. The results showed that there were three valid mediation effect models: (1) “Subjective exercise experience → exercise commitment → low exercise amount”, the confidence interval of this path does not contain 0, indicating that the exercise commitment has a significant mediating effect, in which the standardized effect value: 0.39 × 0.22 = 0.09—accounting for 21.43% of the total effect—and the direct path coefficient of subjective exercise experience to low exercise amount was significant, indicating that exercise commitment plays a partial mediating role in this model. (2) “Subjective exercise experience → exercise commitment → moderate exercise amount”, the confidence interval of this path does not contain 0, indicating that the exercise commitment has a significant mediating effect, in which the standardized effect value: 0.49 × 0.34 = 0.17—accounting for 29.31% of the total effect—and the direct path coefficient of subjective exercise experience to moderate exercise amount was significant, indicating that exercise commitment plays a partial mediating role in this model. (3) “Subjective exercise experience → exercise commitment → high exercise amount”, the confidence interval of this path does not contain 0, indicating that the exercise commitment has a significant mediating effect, in which the standardized effect value: 0.61 × 0.43 = 0.26—accounting for 32.10% of the total effect—and the direct path coefficient of subjective exercise experience to high exercise amount was significant, indicating that exercise commitment plays a partial mediating role in this model. Therefore, hypothesis H4 of this study was confirmed.

## 4. Discussion

### 4.1. Demographic Difference Analysis

Through the difference comparison, it was found that compared with girls, boys have higher levels of exercise commitment and exercise behavior, but there was no significant gender difference in subjective exercise experience. Studies have shown that boys have stronger commitment to exercise than girls and were more likely to form good exercise habits [42], and this may be because boys usually have more persevering personality traits, do not give up and quit easily, boys are livelier and more active, and they are often more enthusiastic and enjoy participating in exercise. Compared with girls, boys will maintain relatively independent, stable, and regular exercise habits in their spare time due to their stronger motivation and desire to exercise [43], which was more likely to increase exercise behavior. Meanwhile, due to the influence of traditional social consciousness and cultural thinking, girls were more introverted and casual in exercise, and it was also difficult to maintain a regular and stable exercise behavior [44]. However, different from previous studies, the subjective exercise experience of boys and girls in this study tends to be consistent, which indicates that during physical exercise, they will feel a relatively similar positive or negative exercise experience, and they could all obtain similar positive exercise experiences, such as pleasure, happiness, and satisfaction through physical exercise, or negative exercise experiences, such as exhaustion and boredom. We speculate that this was affected by factors such as location, exercise environment, and equipment, and both boys and girls in this study belong to the same region, so there is little difference. In terms of age, this study found that the exercise behavior of children and adolescents aged 9–12 was significantly higher than that of children and adolescents aged 13–15, and we speculated that this is because the active nature of younger people is stronger than that of older people, given that they are more willing to participate in physical exercise and enjoy the fun of exercise. Furthermore, the 13–15-year-olds were at a turning point in their personality development, and their academic pressure and tasks were heavier, so their exercise behavior may be reduced. However, the phenomenon that there was no significant age difference in subjective exercise experience and exercise commitment could be explained by the fact that people in these two age groups are minors and their personality characteristics were not fully developed and were relatively close, and in the process of participating in exercise, they felt a more consistent emotional experience of exhaustion, joy, or fun, so their exercise commitment has not yet been significantly differentiated.

### 4.2. The Direct Effect of Subjective Exercise Experience on Exercise Behavior

This study found that there was a significant positive correlation between the subjective exercise experience and exercise behavior of children and adolescents, which showed that subjective exercise experience could directly and positively predict exercise behavior, which could explain 32.49% of the variance. This indicated that subjective exercise experience could influence children and adolescents’ exercise behavior to a greater extent, which showed that the better the exercise experience, the richer the exercise behavior, and vice versa—where the exercise behavior will be restricted. Previous studies have shown that subjective exercise experience could effectively predict individual exercise behavior, in which those with a positive subjective experience were more able to practice physical exercise repeatedly, while those with a negative subjective experience will resist or even withdraw from physical exercise, thereby restricting the frequency of participating in exercise [35,44]. Further research found that a positive exercise experience generated through past exercise can be internalized as fun motivation, which prompts college students to maintain stable exercise behavior in order to meet the need for fun [42]. Since subjective exercise experience was the feeling and experience gained from existing exercise experience, it could enrich the sports cognitive system, improve exercise decision-making ability, and help adolescents to persist in engaging in exercise activities [45]; furthermore, its positive well-being dimension could be used as an important indicator to positively predict individual exercise persistence and directly affect whether individuals will persist in continuing exercise behavior [32,46]. Specifically, when an individual forms a positive experience in previous physical exercise, it is easier to continue to adhere to and maintain a relatively stable exercise behavior, while when an individual has a negative experience, such as psychological distress during previous exercise, they are likely to have a tendency to resist and withdraw from the exercise behavior [32]. Overall, experience was the dependence of cognition and mind on the body, and it was a psychological resource for people to perceive the world, understand the world from the latitude of the body, and use it to change the world [47], wherein having a good workout experience has a direct effect on changing and maintaining the exercise behavior of children and adolescents.

### 4.3. The Mediating Effect of Exercise Commitment between Subjective Exercise Experience and Exercise Behavior

The mediating effect analysis found that the exercise commitment of children and adolescents has a partial mediating effect between subjective exercise experience and exercise behavior, wherein the mediating effect accounts for 37.50% of the total effect, that is, the subjective exercise experience could not only directly affect their exercise behavior, but also indirectly affect their exercise behavior through the mediating effect of exercise commitment. Previous studies have pointed out that exercise commitment was an after-effect of subjective exercise experience [31], wherein the positive exercise experience will fill people with a sense of pleasure and availability, help to stimulate teenagers’ exercise awareness and commitment, and help them to clarify exercise intentions and strengthen their commitment to exercise [30]. This phenomenon could be explained by the decision-making of rational psychological factors on the subject’s behavior, which was inseparable from the reinforcement and activation of irrational factors [48], and as an irrational psychological factor, the subjective exercise experience could effectively predict and strengthen rational psychological factors, such as exercise commitment [32]. Meanwhile, exercise commitment was an intrinsic motivation that affects individual exercise behavior [21], where the theoretical model of exercise commitment could play a certain role in explaining and predicting exercise adherence behavior [49], and maintaining and improving exercise commitment was one of the main factors that promoted individual exercise participation [50]. This showed that a good exercise experience could improve the level of exercise commitment by improving children and adolescents’ exercise fun, personal engagement, and exercise intention, and then play a role in promoting and maintaining exercise behavior. Moreover, the study believes that subjective experience was an influencing factor of exercise commitment, placing it in the leading position of exercise commitment could improve the explanatory power of the cognitive decision model for exercise adherence behavior [45,51]. Moreover, when considering the influence of subjective exercise experience (positive/negative) on individual exercise adherence, under the premise of controlling for workout experience, different exercise commitments may show different exercise adherence behaviors [45]. It can be seen that when children and adolescents experienced a positive exercise experience in the past exercise process, they could internalize it as fun motivation, increase exercise investment and exercise persistence, and then motivate them to maintain a stable exercise behavior to meet the demand for fun.

To further examine the mediating effect of exercise commitment between subjective exercise experience and different exercise amounts, this study divided the exercise behaviors into three levels of low, moderate, and high exercise amounts and tested the mediating effect size in different models. The results showed that exercise commitment had a partial mediating effect between subjective exercise experience and low exercise amount (effect size of 0.09), moderate exercise amount (effect size of 0.17), and high exercise amount (effect size of 0.26), which means that the different exercise amounts of children and adolescents were affected by subjective exercise experience and exercise commitment, but the high exercise amount was the strongest explanation. This showed that there was a linear relationship among the subjective exercise experience, exercise commitment, and exercise behavior of children and adolescents, which means that the higher the level of workout experience, the higher the level of exercise commitment and the richer the exercise behavior. On the contrary, when the workout experience and exercise commitment level were low, individuals often lacked the necessary exercise persistence and fun, thus easily limiting exercise input and corresponding exercise behavior. In previous studies on the relationship between subjective exercise experience, exercise commitment, and exercise behavior, few studies have paid attention to the dose effect of exercise amount, and only some studies found that within a certain range, the higher the level of individual exercise commitment, the higher the frequency of participating in physical exercise, the longer each exercise time is, and the greater the per exercise amount [29]; however, they did not fully consider the effect of exercise experience on exercise commitment and exercise amount. It should be seen that the positive perception and experience of existing exercise could enrich the individual’s conscious thinking or cognitive system of exercise and become an information clue for future behavioral decision-making; furthermore, it made individuals who were originally good at devoting their body and mind to exercise activities have the desire and determination to exercise [42], which in turn was more likely to produce more exercise behaviors and higher exercise amounts. In short, higher exercise experience and exercise commitment were favorable factors for improving individual exercise behavior and exercise amount in children and adolescents.

### 4.4. Limitations

This study mainly explored the mediating effect of exercise commitment between children and adolescents’ workout experience and exercise behavior, and further clarified the effect size differences of different exercise amounts, revealing the importance of a good subjective exercise experience and exercise commitment for changing, maintaining, and promoting exercise behaviors in children and adolescents. However, since this study was a cross-sectional study, the results obtained were more subjective and cannot draw deeper relationships among variables, therefore, longitudinal experimental research should be combined in the future to better reveal the causal relationship among variables. Meanwhile, this study mainly examined the mediating effect of exercise commitment between subjective exercise experience and exercise behavior of different exercise amounts, whether there were other mediating or moderating variables still needs direct evidence from follow-up studies. In addition, this study conducted a model analysis of children and adolescents as a whole, and future research can be divided into age groups to better reveal the age-group differences of them.

## 5. Conclusions

Firstly, there were no significant gender differences in the subjective exercise experience of children and adolescents in this study, but the exercise commitment and exercise behavior of boys were significantly higher than those of girls. Meanwhile, there was no significant age difference in subjective exercise experience and exercise commitment, but the exercise behavior of children and adolescents aged 9–12 was significantly higher than that of children and adolescents aged 13–15. Secondly, there was a significant correlation among children and adolescents’ subjective exercise experience, exercise commitment, and exercise behavior, and the subjective exercise experience could directly and positively predict exercise commitment and exercise behavior—and exercise commitment could also directly and positively predict exercise behavior. Finally, the exercise commitment of children and adolescents played a partial mediating effect between subjective exercise experience and exercise behavior, and both have a mediating effect between subjective exercise experience and different exercise amounts—where the mediating effect was the strongest between subjective exercise experience and high exercise amount.

## Figures and Tables

**Figure 1 ijerph-19-10829-f001:**
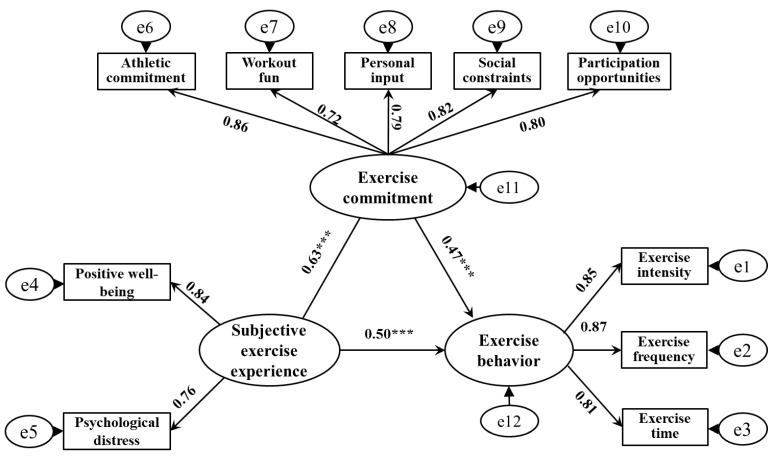
The mediating effect of exercise commitment between subjective exercise experience and exercise behavior. *** *p* < 0.001.

**Figure 2 ijerph-19-10829-f002:**
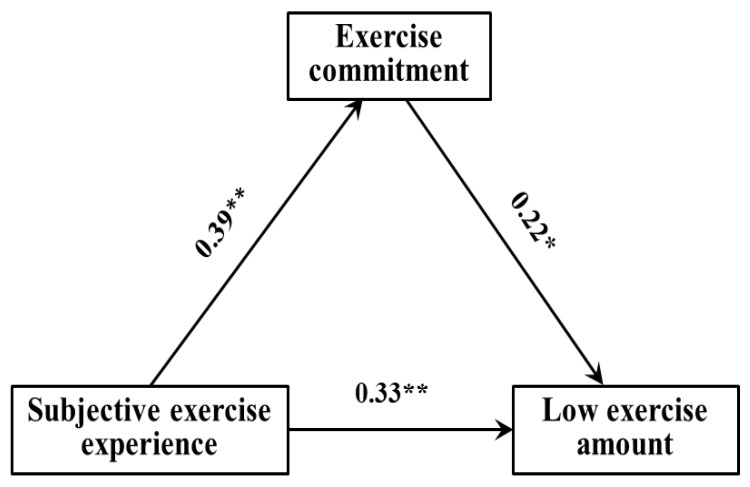
The path model of exercise commitment between subjective exercise experience and low exercise amount. * *p* < 0.05; ** *p* < 0.01.

**Figure 3 ijerph-19-10829-f003:**
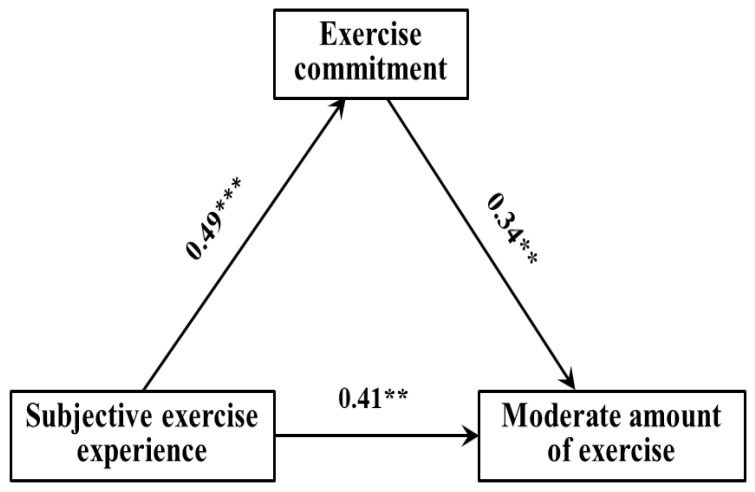
The path model of exercise commitment between subjective exercise experience and moderate exercise amount. ** *p* < 0.01; *** *p* < 0.001.

**Figure 4 ijerph-19-10829-f004:**
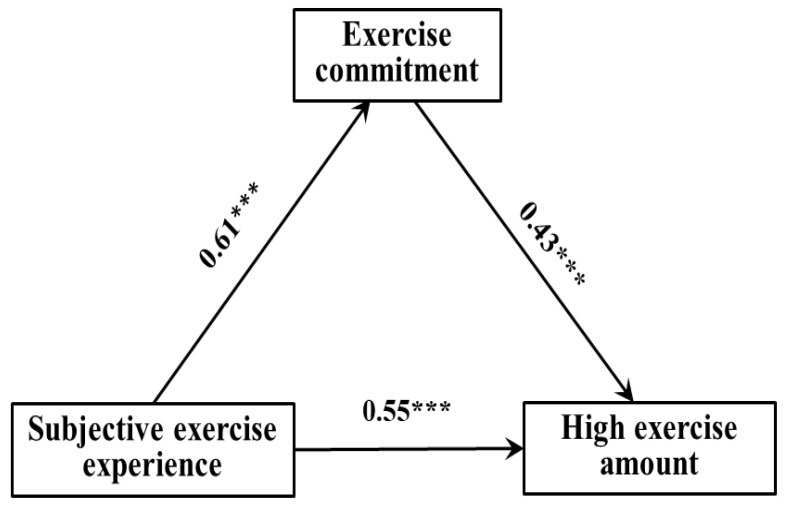
The path model of exercise commitment between subjective exercise experience and high exercise amount. *** *p* < 0.001.

**Table 1 ijerph-19-10829-t001:** Factor extraction and reliability analysis of the three measurement scales.

Scale	KMO and Bartlett Spherical Test	Dimension	Items	Characteristic Root	Variance Explained (%)	Cumulative Explained Variance(%)	Cronbach’s α
SEES	KMO = 0.90 (*p* < 0.001)	Positive well-being	4	3.26	35.17	35.17	0.85
		Psychological distress	4	1.89	16.20	51.37	0.82
ECC	KMO = 0.87 (*p* < 0.001)	Exercise commitment	3	6.83	24.25	24.25	0.86
		Exercise fun	3	4.90	15.18	39.43	0.78
		Personal investment	3	3.55	9.33	48.76	0.80
		Social constraints	3	1.83	6.76	55.52	0.83
		Participation opportunities	3	1.12	4.19	59.71	0.80
PARS-3	KMO = 0.85 (*p* < 0.001)	Amount of physical activity	3	4.03	54.26	54.26	0.82

**Table 2 ijerph-19-10829-t002:** Differences in current status of children and adolescents’ subjective exercise experience, exercise commitment, and exercise behavior.

Category	Variable	Subjective Exercise Experience	Exercise Commitment	Exercise Behavior
Gender	boys	27.89 ± 4.41	49.97 ± 9.03	29.56 ± 6.80
	girls	28.73 ± 5.37	45.08 ± 8.12	25.33 ± 5.39
	t	1.52	3.55 *	3.37 *
Age	9–12	28.57 ± 5.38	48.37 ± 8.11	29.18 ± 6.21
	13–15	28.11 ± 5.09	46.62 ± 7.19	25.71 ± 5.57
	t	0.87	2.45	3.21 *

Note: * means *p* < 0.05.

**Table 3 ijerph-19-10829-t003:** Correlation of the subjective exercise experience, exercise commitment, and exercise behavior.

Variable	M	SD	1	2	3
Subjective exercise experience (1)	28.31	4.89	1	1	1
Exercise commitment (2)	47.52	8.58	0.63 ***		
Exercise behavior (3)	27.45	6.10	0.57 ***	0.52 ***	

Note: *** means *p* < 0.001.

**Table 4 ijerph-19-10829-t004:** Linear regression analysis among the subjective exercise experience, exercise commitment, and exercise behavior.

Variable	Exercise Commitment			Exercise Behavior		
	*β*	R^2^	95%CI	*β*	R^2^	95%CI
Subjective exercise experience	0.63 ***	0.40	(0.59, 0.67)	0.57 ***	0.32	(0.51, 0.60)
Exercise commitment				0.52 ***	0.27	(0.49, 0.58)

Note: *** means *p* < 0.001.

**Table 5 ijerph-19-10829-t005:** Decomposition of path effect of subjective exercise experience on exercise behavior.

Effect Category	Standardized Effect Size	Ratio of Total Effect	Bootstrap *SE*
Total effect	0.80	100%	0.01
Direct effect	0.50	62.50%	0.02
Indirect effect	0.63 × 0.47 = 0.30	37.50%	0.03

**Table 6 ijerph-19-10829-t006:** The mediating effect of exercise commitment between the subjective exercise experience and low exercise amount.

Variable	Exercise Commitment			Low Exercise Amount		
	t	*β*	95%CI	t	*β*	95%CI
Subjective exercise experience Exercise commitment	3.61	0.39 **	(0.35, 0.41)	3.27 2.14	0.33 ** 0.22 *	(0.25, 0.40) (0.16, 0.31)
R^2^	0.15			0.20		
F	4.20 **			5.18 **		

Note: * *p* < 0.05, ** *p* < 0.01.

**Table 7 ijerph-19-10829-t007:** The mediating effect of exercise commitment between subjective exercise experience and moderate exercise amount.

Variable	Exercise Commitment			Moderate Exercise Amount		
	t	*β*	95%CI	t	*β*	95%CI
Subjective exercise experience Exercise commitment	4.93	0.49 ***	(0.45, 0.62)	3.30 4.25	0.34 ** 0.41 **	(0.31, 0.39) (0.36, 0.47)
R^2^	0.24			0.28		
F	7.11 ***			7.67 **		

Note: ** *p* < 0.01, *** *p* < 0.001.

**Table 8 ijerph-19-10829-t008:** The mediating effect of exercise commitment between subjective exercise experience and high exercise amount.

Variable	Exercise Commitment			High Exercise Amount		
	t	*β*	95%CI	t	*β*	95%CI
Subjective exercise experience Exercise commitment	6.25	0.61 ***	(0.52, 0.67)	4.67 5.09	0.43 *** 0.55 ***	(0.38, 0.49) (0.51, 0.62)
R^2^	0.37			0.44		
F	9.58 ***			10.13 ***		

Note: *** *p* < 0.001.

## Data Availability

Not applicable.

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
