# Peer review of "The Effect of Subjective Exercise Experience on Exercise Behavior and Amount of Exercise in Children and Adolescents: The Mediating Effect of Exercise Commitment"

_ijerph, 2022, doi:10.3390/ijerph191710829_

Round 1
Reviewer 1 Report
Overall this is a well-written paper with an interesting result in the sport area.
INTRODUCTION
The introduction provides sufficient background information for readers to understand the problem, however, the authors should clarify relation between exercise experience and exercise behavior. This part would improve this section.
Motivations for this study are more than clear. The objectives are clearly defined at the Introduction and the argumentation in this last part was concise and clarifying.
METHODS
The experimental approach is appropriate for the aim of the study.
This section is well described and allows to replicate the study.
RESULTS
Results paragraphs include more relevant and extended data.
All of the tables include specific, well-developed statistic.
DISCUSSION
All possible interpretations of the data considered are consistent.
Practical application section would improve this point
Limitations are well established and comprehensive.
The conclusions have coherence with the initial aims, in addition, they are well established and according to the present discussion.
LITERATURE CITED
The literature cited is relevant to the study.
Author Response
INTRODUCTION
The introduction provides sufficient background information for readers to understand the problem, however, the authors should clarify relation between exercise experience and exercise behavior. This part would improve this section.
Re: Thanks for your suggestion, we have made targeted revisions to the "Introduction", please check the manuscript!
Motivations for this study are more than clear. The objectives are clearly defined at the Introduction and the argumentation in this last part was concise and clarifying.
Re: Thanks for your recognition.
METHODS
The experimental approach is appropriate for the aim of the study.
This section is well described and allows to replicate the study.
RESULTS
Results paragraphs include more relevant and extended data.
All of the tables include specific, well-developed statistic.
DISCUSSION
All possible interpretations of the data considered are consistent.
Practical application section would improve this point
Limitations are well established and comprehensive.
The conclusions have coherence with the initial aims, in addition, they are well established and according to the present discussion.
Re: Thank you so much for your recognition and encouragement.

Reviewer 2 Report
GENERAL COMMENTS
The aim of this paper was to examine the effect of subjective exercise experience on exercise behaviour among adolescents in Chongqing, China. Although this article addresses an interesting topic, many issues should be addressed before publication.
- Why need children and adolescents? Suggest using one only.
- Major issue – all the three questionnaires are not validated for children, especially those aged 9-12 years.
- Next, the grouping is also inconsistent, for example, 9-12 years and 12-15 years. So, if a child comes with 12 years old, will he or she be in group A or B?
SPECIFIC COMMENTS
INTRODUCTION
The introduction needs major revision and clarification. First, the aim of the study is not clear. The research article should answer a specific research question, which is lacking in this study. Also, the way the authors build up their introduction does not lead to the research question. Although much of the necessary information regarding the background is already written down, the authors should re-structure their introduction, explaining why their research is important. The authors described some studies on exercise experience and exercise behaviour separately, but the discussion of both together is very shallow.
METHODS
As it stands, it is not possible to replicate their study. The authors should justify why the participants aged 9-12 years and 12-15 years were recruited? Both age group are different. What is the inclusion and exclusion criteria? What is the sample size calculation?
All three questionnaires are not validated for children, especially those aged 9-12 years. If it is not suitable for children, it should not be used?
SPSS version 21 does not exist anymore unless the authors are using non-licensed software?
RESULTS
The results section needs minor revision. The authors should add how they gathered all the information. Probably, including a research question would help the authors to structure their results. I think the authors put too much information in the tables, making the entire manuscript hard to follow.
DISCUSSION
In the discussion section, the authors should further discuss their findings and the implication of these findings. They should also discuss their findings in more depth. The authors also discuss many topics that are not related to the results. Suggest to tighten up the discussion.
The limitation is also not well thought out. Please revise. Future research direction is missing.
The conclusion should be in one paragraph and not encourage to use point form.
Thank you.
Author Response
GENERAL COMMENTS
The aim of this paper was to examine the effect of subjective exercise experience on exercise behaviour among adolescents in Chongqing, China. Although this article addresses an interesting topic, many issues should be addressed before publication.
- Why need children and adolescents? Suggest using one only.
Re: Thanks for your suggestion. We have checked a lot of information and found that many studies also combine children and adolescents into a whole for research, and do not make detailed distinctions. for example, â‘ Feldstein, L.R., et al.,2020. Multisystem Inflammatory Syndrome in US Children and Adolescents. New England Journal of Medicine, 383(4),334-346. doi:10.1056/NEJMoa2021680; â‘¡Tarp, J., et al.,2018. Physical activity intensity, bout-duration, and cardiometabolic risk markers in children and adolescents. International Journal of Obesity, 42,9,1639-1650.doi: 10.1038/s41366-018-0152-8; â‘¢Ma, S.K., et al., 2022. Bout Characteristics of MVPA and its Relationship with Physical Fitness in Children and Adolescents. China Sport Science,42(04),43-49+97.doi:10.16469/j.css.202204005; â‘£Hong, J.J., et al., 2021. Influence of Sports Environment on Children and Adolescents’Exercise Behavior: Intermediary Effect of Sports Commitment. China Sport Science and Technology,1-7.doi: 10.16470/j.csst.2021046…In addition, this study is a preliminary study to explore the mediating effect of exercise commitment between subjective exercise experience and exercise behavior in a large group of children and adolescents. In the follow-up study, we will further distinguish children and adolescents. In-depth exploration can even be extended to other age groups.
- Major issue – all the three questionnaires are not validated for children, especially those aged 9-12 years.
Re: First, for the "Subjective Exercise Experience Scale", the revised SEES developed by Mcauley (1994) proved to have good reliability and validity, while Markland et al. (1997) verified SEES in children and confirmed that the tool also Suitable for children. Second, for the Physical Exercise Rating Scale, Li et al. (2020, doi: 10.16128/j.cnki.1005-3611.2020.06.045) validated PARS-3 in children and confirmed that the tool is suitable for Chinese children. Regarding the Exercise Commitment Scale, although no studies have been performed on children, this study has further provided reliability and validity analysis results including test-retest, indicating that all three scales are suitable for children.
- Next, the grouping is also inconsistent, for example, 9-12 years and 12-15 years. So, if a child comes with 12 years old, will he or she be in group A or B?
Re: We are very sorry, this is a misunderstanding caused by our misrepresentation, we have revised it to 9-12 years and 13-15 years in the manuscript.
SPECIFIC COMMENTS
INTRODUCTION
The introduction needs major revision and clarification. First, the aim of the study is not clear. The research article should answer a specific research question, which is lacking in this study. Also, the way the authors build up their introduction does not lead to the research question. Although much of the necessary information regarding the background is already written down, the authors should re-structure their introduction, explaining why their research is important. The authors described some studies on exercise experience and exercise behaviour separately, but the discussion of both together is very shallow.
Re: Thanks for your suggestion, we have revised and clarified the "Introduction" section, please consult the manuscript.
METHODS
As it stands, it is not possible to replicate their study. The authors should justify why the participants aged 9-12 years and 12-15 years were recruited? Both age group are different. What is the inclusion and exclusion criteria? What is the sample size calculation?
Re: We refer to a lot of research on children and adolescents and find that 3-17 years old belong to this group. In this study, considering that filling in the questionnaire requires a certain level of language comprehension and cognitive ability, the younger age groups were excluded, and considering that the age span should not be too large, the groups over 15 years old were excluded. Therefore, this study finally selected children and adolescents aged 9-15 as the survey objects, and divided them into two age groups: 9-12 years old and 13-15 years old (in China, 9-12 years old corresponds to the upper grade of primary school, that is, Grades 4-6; and 13-15 years old correspond to junior high schools, namely the first, second and third grades). Regarding the sample size, the sample survey method was used in this study to survey 7 elementary and junior high schools at a ratio of about 1:40. Inclusion and exclusion criteria have been supplemented in the manuscript, please see "1.1 Participants".
All three questionnaires are not validated for children, especially those aged 9-12 years. If it is not suitable for children, it should not be used?
Re: Thanks for your suggestion, we've answered it in the question above.
SPSS version 21 does not exist anymore unless the authors are using non-licensed software?
Re: We use the Chinese version of SPSS 21.0, and its results are completely consistent with SPSS 13.0, and it is recognized in China, and many studies use this version.
RESULTS
The results section needs minor revision. The authors should add how they gathered all the information. Probably, including a research question would help the authors to structure their results. I think the authors put too much information in the tables, making the entire manuscript hard to follow.
Re: Thanks for your suggestion, we have sorted out some of the "results" and simplified some of the tables, please refer to the manuscript.
DISCUSSION
In the discussion section, the authors should further discuss their findings and the implication of these findings. They should also discuss their findings in more depth. The authors also discuss many topics that are not related to the results. Suggest to tighten up the discussion.
Re: Thanks for your suggestion, we have made further enrichments and corrections to the Discussion section, please check out the manuscript!
The limitation is also not well thought out. Please revise. Future research direction is missing.
Re: Thanks for your suggestion, we have made further enrichments and corrections to "limitation", please check the manuscript!
The conclusion should be in one paragraph and not encourage to use point form.
Re: Thanks for your suggestion, we have fixed the "conclusion".
Thank you.
Reviewer 3 Report
This manuscript does an interesting job demonstrating the influencing factors that restrict the exercise behaviour of children and adolescents (boys and girls). The analytical tools used by the author made this article more compile information and important data that can be added to the field of exercise science.
Minor revisions
1. About the research tools
a. Summarize the Subjective Exercise Experience Scale 16 (SEES), Exercise Commitment Scale (ECC), and Physical Exercise Rating Scale (PARS-3) and added to the most important and relevant data used for inferring the conclusions and made tables.
b. Some minor syntaxis and grammar corrections.
About the paper
The title doesn’t reflect the actual research conclusions it should be modified accordingly.
The summary, the word subjective exercise experience and not significant is too repetitive. It should be rewritten and corrected for running sentences.
Are there any explanations for why this is not significant between gender differences in subjective exercise experience since previous research shows significance?
How does the author correct the questionaries for bias? How does this was performed?
Is there any social-economic analysis that can impact this analysis?
Quality of life was assessed among the participants. Other related factors that can affect the subjective exercise experience on exercise behaviour?
Any analysis of parents promoting exercise behaviour? Time spent with the child and doing sport or exercise with the child. At home or any other exercise place, like playgrounds, parks and so on.
Author Response
This manuscript does an interesting job demonstrating the influencing factors that restrict the exercise behaviour of children and adolescents (boys and girls). The analytical tools used by the author made this article more compile information and important data that can be added to the field of exercise science.
Minor revisions
- About the research tools
- Summarize the Subjective Exercise Experience Scale (SEES), Exercise Commitment Scale (ECC), and Physical Exercise Rating Scale (PARS-3) and added to the most important and relevant data used for inferring the conclusions and made tables.
Re: Thanks for your suggestion, we have supplemented the manuscript accordingly, please refer to section "1.2 Measurement tools".
- Some minor syntaxis and grammar corrections.
Re: Thanks for your suggestion, we have made some corrections.
About the paper
The title doesn’t reflect the actual research conclusions it should be modified accordingly.
Re: Thanks for your suggestion, we have edited the title accordingly.
The summary, the word subjective exercise experience and not significant is too repetitive. It should be rewritten and corrected for running sentences.
Re: Thanks for your suggestion, we have replaced subjective exercise experience and not significant in the manuscript as much as possible, but many places are really difficult to replace, please check the manuscript.
Are there any explanations for why this is not significant between gender differences in subjective exercise experience since previous research shows significance?
Re: Thanks for your suggestion, we have supplemented the corresponding explanation in the manuscript, please consult the Discussion section.
How does the author correct the questionaries for bias? How does this was performed?
Re: The author tested the bias in the questionnaire through the common method bias according to the viewpoint of the predecessors, and the relevant content has been supplemented to "2.1 Common method bias test".
Is there any social-economic analysis that can impact this analysis?
Any analysis of parents promoting exercise behaviour? Time spent with the child and doing sport or exercise with the child. At home or any other exercise place, like playgrounds, parks and so on.
Re: Thank you for your suggestion. This study mainly involves children and adolescents, so we only focus on the effects of gender and age on variables, and control them in the model analysis. As for the influence of factors such as socioeconomics and parents, it would be another good research direction, but it was not covered in this study, and we hope to investigate further in future research, thank you.
Quality of life was assessed among the participants. Other related factors that can affect the subjective exercise experience on exercise behaviour?
Re: Thank you for your suggestion. There should be many factors that affect subjective exercise experience and exercise behavior. However, this study mainly focuses on the relationship between subjective exercise experience, exercise commitment, exercise behavior and exercise volume. It is appropriate to add too many factors, otherwise it will be difficult to explain and present the core issue.
